# Optimized Combination of Spray Painting Trajectory on 3D Entities

**Wei Chen [1,2,*], Xinxin Wang [1], Hao Liu [1], Yang Tang [3] and Junjie Liu [1]**

[1] School of Electronics and Information, Jiangsu University of Science and Technology, Zhenjiang 212003, China; 172030030@stu.just.edu.cn (X.W.); directionod@aliyun.com (H.L.); junjiel_mtr@163.com (J.L.)

[2] School of Automation, Southeast University, Nanjing 210096, China

[3] School of Science, Jiangsu University, Zhenjiang 212013, China; ty800117@ujs.edu.cn

* Correspondence: cwchenwei@aliyun.com

**Abstract:** In this research, a novel method of space spraying trajectory optimization is proposed for 3D entity spraying. According to the particularity of the three-dimensional entity, the finite range model is set up, and the 3D entity is patched by the surface modeling method based on FPAG (flat patch adjacency graph). After planning the spray path on each patch, the variance of the paint thickness of the discrete point and the ideal paint thickness is taken as the objective function and the trajectory on each patch is optimized. The improved GA (genetic algorithm), ACO (ant colony optimization), and PSO (particle swarm optimization) are used to solve the TTOI (tool trajectory optimal integration) problem. The practicability of the three algorithms is verified by simulation experiments. Finally, the trajectory optimization algorithm of the 3D entity spraying robot can improve the spraying efficiency.

**Keywords:** spray painting robot; FPAG; GA; ACO; PSO; TTOI problem

## 1. Introduction

With the development of the social economy and the improvement of life, people have higher requirements for product quality and product appearance. Moreover, some products with higher surface spraying quality, such as furniture, automobiles, and artworks, determine the quality of the product appearance and the competitiveness of products in the market. Therefore, people pay more and more attention to surface spraying technology [1]. The traditional spraying technology is that the spraying workers with spraying guns directly spray the workpiece to be sprayed. During spraying, the paint mist diffuses into the surrounding environment, which not only pollutes the environment, but also seriously harms the physical and mental health of spraying workers. To solve the problems caused by traditional spraying, automatic spraying systems have been developed. As a typical table of spraying automation equipment, a spraying robot has many advantages, such as good uniformity of the coating thickness, high repetitive positioning accuracy, wide applicability, and high efficiency. At the same time, spraying robots can free workers from a toxic, flammable, and explosive working environment [2].

In many fields of industrial production, the surface spraying of the workpiece is an essential step, and the types of workpiece modeling are becoming more and more abundant. During the spraying operation, the workpiece may have multiple spraying surfaces, and the curvature of each spraying surface is different. Usually, a three-dimensional (3D) entity modeling method is needed to form these parts, as shown in the 3D entity diagram in Figure 1. In recent years, due to the wide application of 3D solid modeling in various fields, scholars have made remarkable achievements in the research of 3D

solid modeling [3–7]. The spraying of 3D solid modeling has a certain novelty because the spraying is more interested in the surface shape of the workpiece. For the research of this paper, we use the finite spraying modeling technology and the surface modeling method based on the plane patch adjacency graph (FPAG) for modeling. It should be pointed out that the 3D entities mentioned in this paper are not the same as polyhedra. A polygon is a spatial geometry surrounded by several planar polygons, each of which is a plane. However, each face of a 3D entity can be a free-form curved surface. In this sense, the polyhedron is only a subset of the 3D entity.

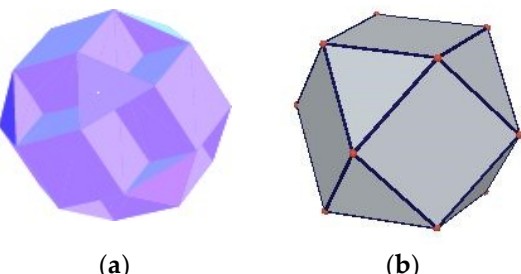

(**a**)　　　　　　　　(**b**)

**Figure 1.** (**a**) 3D entities with concave surfaces. (**b**) 3D entities with convex surfaces.

In addition, research on trajectory optimization for spray painting robots oriented to 3D entities is immature. Existing trajectory optimization methods can only be applied to 3D entities with convex surfaces, as shown in Figure 1b. For a 3D entity with concave surfaces (Figure 1a), the research on the trajectory optimization for the robot in this field is still a blank as the shape of the entity is complex and the robot is required to be extremely flexible in automatic spray painting operation. The trajectory optimization method for the spray painting robot introduced in this paper can only be applied to 3D entities with convex surfaces.

The idea of trajectory optimization for a spray painting robot oriented to a 3D entity is: First, a simple mathematical model of the paint deposition rate is established by the experimental method and the 3D entity is sliced by using the FPAG surface modeling method [8–10]. Secondly, after planning the painting path on each patch, the spray painting trajectory is optimized on each patch with the objective function of the variance of the paint thickness at discrete points and the ideal paint thickness. Finally, the spray painting trajectories on the individual patches are optimized, and the optimized trajectories of the spray painting robot on the 3D entities are formed eventually.

## 2. The Establishment of the Mathematical Model

The spraying mathematical model mainly includes the position and direction of the end-effector and the paint deposition rate model.

The establishment of the paint deposition rate model is an important problem in trajectory optimization for spray painting robots. Considering that the expressions of these models are rather complex, it is not applicable to build the mathematical model of the paint deposition rate on 3D entities. Therefore, in this paper, the specific mathematical expression of the finite range model takes the following factors into account: The distance and direction from the gun to the working surface, the curvature of the workpiece surface, and the angle of the paint gun (i.e., the cone angle corresponding to the paint flow).

Before building a finite-scope model, we made the following assumptions: The paint particles sprayed by the paint gun form a cone in space. Suppose the opening angle, $\varphi$, is half of the angle, $\varphi$, and the opening angle, $\varphi < 90°$. The definition of the opening angle, $\varphi$, can be seen in Figure 2.

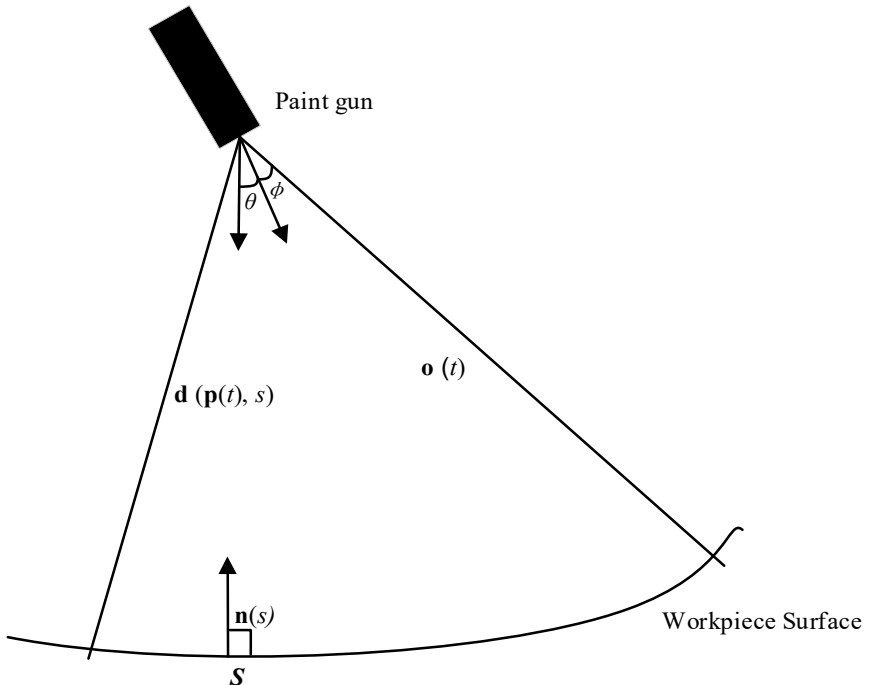

**Figure 2.** The finite range model.

Considering that the amount of paint to be obtained is smaller when the distance, $L(L > 0)$, from the point, $s(x, y, z) \in S$, on the surface to the gun increases. Additionally, the amount of paint obtained at the points on the surface is also decreased when the angle, $\theta(\theta < \varphi)$, is gradually increasing. The amount of the paint obtained at the point, $s$, on the workpiece surface can be expressed by the following equation:

$$\frac{c(\theta, \phi)}{L^2} \tag{1}$$

Among which:

$$c(\theta, \varphi) \begin{cases} > 0, & \theta < \varphi \\ = 0, & \theta \geq \varphi \end{cases} \tag{2}$$

$$L = \sqrt{(x - \mathbf{p}_x(t))^2 + (y - \mathbf{p}_y(t))^2 + (z - \mathbf{p}_z(t))^2} \tag{3}$$

$\mathbf{p}_x$, $\mathbf{p}_y$, and $\mathbf{p}_z$ express, respectively, the coordinates of the gun on the X-axis, Y-axis, and Z-axis. Assuming that the workpiece surface is a curved surface, the paint deposition rate of a point, $s(x, y, z) \in S$, on the surface is proportional to the inner product of the following two vectors (shown in Figure 2): First, the unit normal vector, $n(s)$, of the point. Second, the unit direction vector, $d(p(t), s)$, of the gun and the point, $s$. The function, $d(p(t), s)$, is defined as follows:

$$d(p(t), s) = \frac{(x - \mathbf{p}_x(t))\mathbf{i} + \left(y - \mathbf{p}_y(t)\right)\mathbf{j} + (z - \mathbf{p}_z(t))\mathbf{k}}{L} \tag{4}$$

where, $\mathbf{i}$, $\mathbf{j}$, and $\mathbf{k}$ denote unit vectors in the positive direction of the X, Y, and Z axis, respectively.

Based on the above assumptions, we can derive the paint deposition rate function of a point, $s$, on the surface of the workpiece:

$$\dot{f}(s, p(t), t) = \left( \frac{c(\theta, \varphi)}{(x - \mathbf{p}_x(t))^2 + (y - \mathbf{p}_y(t))^2 + (z - \mathbf{p}_z(t))^2} \right) \cdot d(p(t), s) \cdot n(s) \tag{5}$$

The variable, $\theta$, in Equation (5) is not significant. In the equation, $\theta$ is the angle of the gun and a point, *s*, on the workpiece surface and the axis of the gun. It is related to the coordinates of the point, *s*, on the workpiece surface, the position, $\boldsymbol{p}(t)$, and direction, $\boldsymbol{o}(t)$, of the paint gun, which can be defined as:

$$\theta = \cos^{-1}(\boldsymbol{d}(\boldsymbol{p}(t), s) \cdot \boldsymbol{o}(t)) \tag{6}$$

The selection of function, $c(\theta, \varphi)$, depends on the basic spray characteristics of the gun and some parameter settings, such as the air pressure of the gun and the velocity of the paint flow. In the usual case, the function, $c(\theta, \varphi)$, reaches its maximum when $\theta = 0$ (that is, it just below the paint gun). When $\theta \to \varphi$, $c(\theta, \varphi) \to 0$. The following gives a concrete model of function, $c(\theta, \varphi)$:

$$c(\theta, \varphi) = \begin{cases} \alpha \dfrac{\cos(\theta) - \cos(\varphi)}{(1 - \cos(\varphi))^2} & \theta \leq \varphi \\ 0 & otherwise \end{cases} \tag{7}$$

The painting velocity of the paint gun and air pressure can be adjusted by changing the values of the parameters, $\alpha$ and $\varphi$, in the formula above, respectively. It can also be seen from the formula above that the values of the parameters, $\alpha$ and $\varphi$, are related to the maximum of the function, $c(\theta, \varphi)$.

The finite-range model here is circular on the plane corresponding to the paint distribution model. When the workpiece surface has a certain curvature, the paint distribution should become oval. For surfaces with small curvatures, the paint distribution, which is actually elliptical, can be approximated as a circular. In addition, the paint deposition rate model also illustrates that the total amount of paint sprayed to the workpiece surface (the total amount of paint sprayed from the gun) has nothing to do with the surface shape of the workpiece and the distance from the gun to the workpiece surface, which is in line with the actual situation.

## 3. Segmentation for 3D Entity

To obtain the optimal trajectory on the 3D solid surface, the first step is to model the workpiece surface. Second, due to the particularity of the spraying surface, trajectory optimization of the spraying robot is relatively difficult in a practical application. To obtain a good spraying effect, the finite range model modeling method is adopted. The third step is to use the FPAG surface modeling method to simplify the plane patch adjacency graph. As shown in Figure 3, the specific steps are as follows: (1) Divide the triangular grid of the 3D solid surface. (2) Set the maximum normal vector threshold and connect the triangle surface into a smaller flat area according to the triangle connection algorithm. (3) Each patch is approximately planar, and at least one side of each patch is part of the 3D solid ridge.

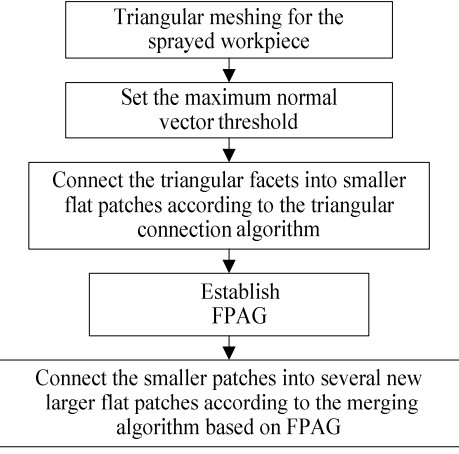

**Figure 3.** Step diagram of the surface modeling method based on the flat patch adjacency graph (FPAG).

## 4. Trajectory Optimization on Each Patch

The geometric properties of 3D entity surfaces are more complex compared with two-dimensional planar surfaces and regular curved surfaces. Therefore, to simplify the problem, we will describe a relatively simple and practical trajectory optimization method in the following. This method is fast and the process is simple. It can meet the actual needs completely.

The trajectory of a spray painting robot mainly consists of two factors: Path and velocity. In the spray painting process, the painting path can be obtained by determining the width of the overlapping areas formed by two paint strokes. Therefore, to determine the trajectory of a spray painting robot, we only need to determine the velocity of the paint gun and the width of the overlapping area formed by two paint strokes. $x$ represents the distance from a point, $s$, to the first path in the painting radius, $d$ represents the width of the overlapping area formed by two paint strokes, $R$ represents the distance from the surface point to the spray direction, $o(t)$, O is the TCP (tool center point). Then, the paint thickness at point, $s$, is:

$$q_s(x) = \begin{cases} lq_1(x) & 0 \le x \le R - d \\ q_1(x) + q_2(x) & R - d < x \le R \\ q_2(x) & R < x \le 2R - d \end{cases} \tag{8}$$

$q_1(x)$ and $q_2(x)$ are the paint thickness at point $s$ when spray painting on two adjacent paths, respectively. The formulas of $q_1(x)$ and $q_2(x)$ are:

$$q_1(x) = 2\int_0^{t_1} f(r_1)dt, \ 0 \le x \le R \tag{9}$$

$$q_2(x) = 2\int_0^{t_2} f(r_2)dt, \ R - d \le x \le 2R - d \tag{10}$$

among which, $t_1 = \sqrt{R^2 - x^2}/v$, $t_2 = \sqrt{R^2 - (2R - d - x)^2}/v$, $r_1 = \sqrt{(vt)^2 + x^2}$, $r_2 = \sqrt{(vt)^2 + (2R - d - x)^2}$. $t_1$ and $t_2$ represent half of the spray painting time that the gun paints in two adjacent painting paths at point $s$, respectively. $r_1$ and $r_2$ represent the distance from point $s$ to the central projection point of TCP in two adjacent painting paths, respectively, $t$ is the time that the gun moves from point O to point $s'$. $s'$ is the projection of the point, $s$, on the painting path. The following expression can be obtained from (9) and (10):

$$q_s(x, d, v) = \frac{1}{v}J(x, d) \tag{11}$$

where, $J$ is a function of $x$ and $d$. To make the paint thickness on the surface as uniform as possible, the difference between the actual paint thickness and the variance of ideal paint thickness at point $s$ are taken as the optimization objective function:

$$\min_{d\in[0,R],v} E_1(d, v) = \int_0^{2R-d} (q_d - q_s(x, d, v))^2 dx \tag{12}$$

In the equation above, $q_d$ is the ideal paint thickness. Since the maximum paint thickness, $q_{max}$, and the minimum paint thickness, $q_{min}$, determine the uniformity of the paint thickness on the workpiece surface, and $q_{max}$ and $q_{min}$ also need to be optimized:

$$\min_{d\in[0,R],v} E_2(d, v) = (q_{max} - q_d)^2 + (q_d - q_{min})^2 \tag{13}$$

It can be obtained from Equations (11)–(13) that:

$$\min_{d\in[0,R],v} E(d, v) = \frac{1}{2R - d}E_1(d, v) + E_2(d, v) \tag{14}$$

Additionally, from Equations (9) and (10), the expressions of the maximum paint thickness and minimum paint thickness can be described as:

$$q_{max} = \frac{1}{v} J_{max}(d) \tag{15}$$

$$q_{min} = \frac{1}{v} J_{min}(d) \tag{16}$$

Let $\frac{\partial E(d,v)}{\partial v} = 0$, it can be obtained from Equations (11), (14)–(16) that:

$$v = \frac{\frac{1}{2R-d} \int_0^{2R-d} J^2(x,d)dx - J_{max}^2(d) - J_{min}^2(d)}{q_d[\frac{1}{2R-d} \int_0^{2R-d} J(x,d)dx + J_{max}(d) + J_{min}(d)]} \tag{17}$$

It can be seen that the spray painting rate, $v$, can be expressed as a function of the width, $d$, of the paint overlapping area formed by two painting strokes. Therefore, the minimum value of $E(d,v)$ is only related to $d$. The golden section method [11] can be used to obtain the optimization value, $d$, so that the optimized trajectory on each patch can be obtained too.

## 5. Tool Trajectory Optimal Integration on 3D Entity

After the track optimization of each patch, the optimal combination of trajectories connecting each patch should also be considered to speed up the spraying speed of the spraying robot. The first step is to transform and model the TTOI problem. TTOI problem is represented by the Hamiltonian diagram. In the second step, the corresponding optimization algorithm is used to solve the TTOI problem. The third step, through simulation and spray painting experiments, verification, and comparison, identifies the advantages and effectiveness of the algorithm.

### 5.1. The Transformation and Modeling of Tool Trajectory Optimal Integration

As is shown in the Figure 4, the TTOI (tool trajectory optimal integration) on each patch after the 3D entity segmentation is expressed [12]. To make the problem less complicated, the trajectories are considered as an edge. The ultimate purpose of the TTOI problem is to spray patches on the workpiece surface to make the spraying path of the robot the shortest. According to graph theory, a non-directional connection graph, G, is assumed ($V$, $E$, $R$, $\omega$: $E \rightarrow Z^+$), among which $V$ denotes the vertex set, $E$ denotes the edge set, $R$ denotes any subset of $E$, and $\omega$ denotes the weight of the edge (the length of the actual spray path). The problem of TTOI is to find a path passing all edges only once with the shortest distance in graph, G. Similar to the traveling salesman problem (TSP), which is a common problem in the optimization problem, the TTOI problem is also a typical NP (non-polynomial) problem.

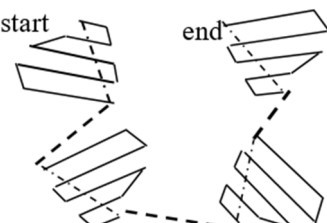

**Figure 4.** Tool trajectory optimal integration on each patch.

Suppose that D = $\{d_{ij}\}$ ($i, j = 1, 2, \ldots, n$), the shortest distance between vertex $i$ and vertex $j$, which are not on the same edge in graph G, the distance between the vertices can be calculated according to the Floyd algorithm. To make the problem less complicated, the TTOI problem can be expressed by the Hamiltonian method. As shown in Figure 5, a vertex is used to represent an edge of the original graph

G to form a complete Hamiltonian [12]: *g* (*VH*, *EH*, *ωH*), among which *VH* denotes the vertex set, *EH* denotes the edge set, and *ωH* denotes the weight of the edge and *ωH*∈D. In the graph, *g*, the weight of each edge is not fixed. Its value is determined by the order of vertices on the same edge in the original graph, G. Suppose that the order of the vertices set, *VH* = {$v_1$, $v_2$ ... ... $v_n$}, in graph g is *T* = ($t_1$, $t_2$ ... ... $t_n$) $t_i$∈*VH* (*i* = 1, 2, ... , *n*), and the TTOI problem can be defined as Equation (18):

$$\min\overline{L} = \sum_{i=1}^{n}\omega_i + \sum_{j=1}^{n-1}\omega_j^{H} \tag{18}$$

where $\omega_i$ is the weight of the edges in the primitive graph, G, corresponding to vertices, $t_1$, $t_2$ ... ... $t_n$, in graph g and $\omega_j^{H}$ denotes the weight of the edge in graph *g*. Since the weight, $\omega_i$, of each edge in the original graph, G, is considered to be a fixed value in this problem, the above optimization problem can be reduced to Equation (12):

$$\min L = \sum_{j=1}^{n-1}\omega_j^{H} \tag{19}$$

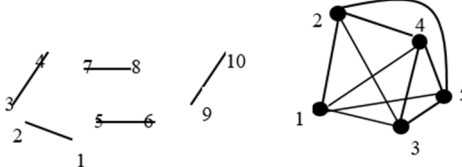

**Figure 5.** Transformation of the original graph, *G*, into the Hamiltonian graph, *g*.

The spraying robot is the most complex one in the control of the industrial robot because of its many parameters. Especially, the trajectory optimization of the spraying robot on the complex surface makes the actual operation difficult. Therefore, finding the arrangement of all vertices in the Hamilton diagram makes the path, L, of the painting robot the shortest, which becomes a TTOI problem. Because of the large number of parameters, spray robots are the most complex control of industrial robots. Especially, the trajectory optimization of the spraying robot on the complex surface makes the actual operation difficult. Therefore, finding the arrangement of all vertices in the Hamilton diagram makes the shortest path, L, of the painting robot a TTOI problem. To solve the TTOI problem, the improved genetic algorithm, ant colony algorithm, and particle swarm optimization (PSO) proposed in this paper can be used to optimize the trajectory of a spraying robot on complex surfaces. In the process of fragmentation, these intelligent algorithms can solve the trajectory optimization problem between patches. For the first time, these intelligent algorithms have been used for spraying complex surfaces. The previous links between patches are random combinations. Finally, the advantages and disadvantages of each algorithm are illustrated by experiments.

*5.2. Solving the TTOI Problem with the Genetic Algorithm*

Genetic algorithm (GA) is a method to search for the optimal solution by simulating the natural evolution process. Therefore, this algorithm has good effects on the NP (non-polynomial) problem in combinatorial optimization and can be used to solve the TTOI problem. According to the particularity of TTOI [13–15], GA needs a special individual code and crossover, mutation, and other genetic manipulation methods.

**(1) Individual code:** The length of the individual code is |*VH*|. Since each vertex in the Hamilton graph represents one edge of the original graph, G, to distinguish the start and end points of each edge. The individual code contains the binary code, $P_{si}$, representing the direction of each edge in the original graph, G.

**(2) Fitness function:** The values of fitness function are used to determine which individuals can enter the next round of evolution and which individuals need to be removed from the population.

To facilitate the selection operation in the genetic algorithm, the optimization of the minimum value is usually converted to optimization of the maximum value, and the fitness function can be taken as: $F = U - L$, where $U$ should be selected as an appropriate number, to make the fitness of all individuals positive. In the process of population evolution, to select the individuals with high fitness, the population size is maintained as the value, $P_{size}$. According to the fitness function rule, the $P_{size}$ individuals with the highest fitness are passed to the next generation.

**(3) Crossover:** Crossover is the process of exchanging the partial codes between two individuals with a certain probability to generate new individuals. Here, order crossover (OX) is used on $P_i$ while two-point crossover is used on $P_{si}$. OX ensures that the original order of each vertex is almost the same when the effective sequence of the individual itinerary is modified [16]. The main idea of OX is: A conventional two-point crossover is performed, followed by an effective sequence modification of the individual itinerary. When modifying, the original relative access order of each point should be maintained as much as possible. Basic steps of OX are as follows:

(a) In the individual code strings, $P_x$ and $P_y$, representing the spray painting order, the positions after the two loci, $i$ and $j$, are randomly selected as the intersection. That is, each locus between the $(i + 1)$-th locus and the $j$-th locus is defined as an intersection area, and the contents of the intersection region are respectively memorized to $W_x$ and $W_y$.

(b) According to the mapping relation in the intersection area, find all $P_{xq} - P_{xq}$ ($p = i + 1, i + 2,$ $\ldots, j$) loci $q$ in the individual $P_x$ and set them as vacancies. Find all $P_{xq} - P_{xq}$ ($p = i + 1, i + 2, \ldots, j$) loci, $r$, in the individual, $P_y$, and set them as vacancies.

(c) The individuals, $P_x$, $P_y$, are left-shifted circularly until the first vacancy in the code string is moved to the left end of the intersection area. Then, all the vacancies are concentrated in the intersection area, and the original gene values in the intersection area are sequentially moved backward.

(d) Exchange the content in $W_x$ and $W_y$ and put them into the intersection area of individual $P_x$, $P_y$. The result is a new spray painting order.

**(4) Mutation operation:** $P_i$ is subjected to inversion mutation to generate a new individual. A basic variation is applied to $P_{si}$, where one or more loci are randomly selected for individual code and the gene values of these loci are inverted.

Thus, the genetic algorithm of the TTOI problem is shown in Figure 6.

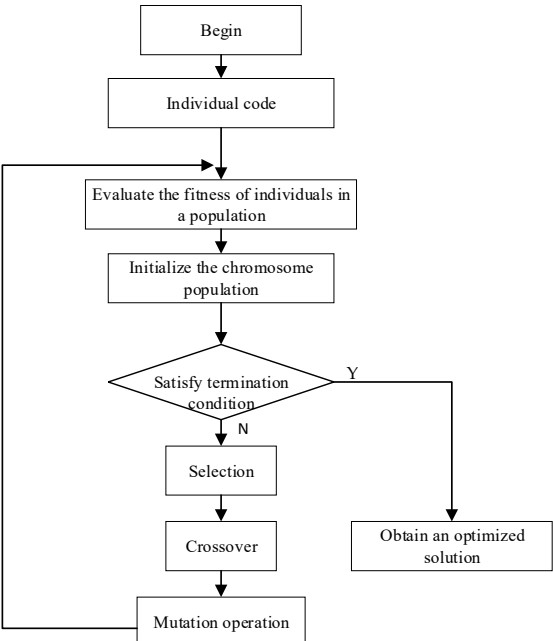

**Figure 6.** Flow chart of the GA (Genetic algorithm) of the TTOI (tool trajectory optimal integration) problem.

Taking 3D entities as spray objects, simulation experiments were carried out using genetic algorithm programming to verify the effectiveness of the TTOI problem. According to the segmentation method of the 3D entity, a 3D entity is divided into seven patches; that is, the individual code, $P_i$ and $P_{si}$, are seven bits in the genetic algorithm. The parameters of the algorithm are as follows: Population size, $P_{size}$ = 100; crossover probability, $x_{rate}$ = 0.20; mutation probability, $m_{rate}$ = 0.05; and the maximum number of evolutionary generation, $T$ = 100. The corresponding evolutionary processes of different solutions are shown in Figure 7. As can be seen from the figure, the value of the objective function of the optimal individual decreases monotonously with the evolution process and eventually tends to be constant. After about 70 generations of evolution, the average fitness remains stable and the algorithm converges.

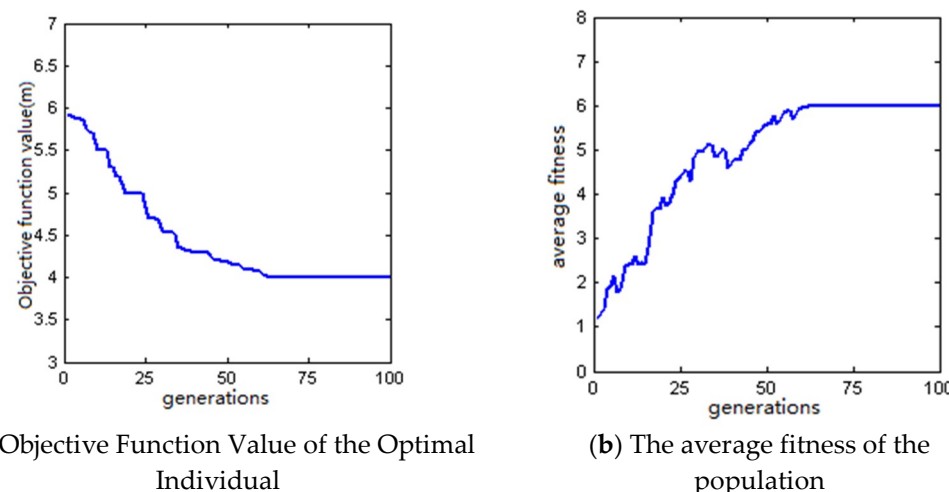

(**a**) Objective Function Value of the Optimal Individual

(**b**) The average fitness of the population

**Figure 7.** The simulation result of solving the TTOI problem with GA.

### 5.3. Solving the TTOI Problem with the Ant Colony Algorithm

Ant colony optimization (ACO) is a probabilistic intelligent algorithm for finding the optimal path. It originated from the behavior of ants to find the path in the process of searching for food. It has strong anti-interference ability, strong compatibility, and other characteristics. The algorithm initializes the following individual information: Not visited vertices (NVV), not visited edges (NVE), visited edges (VE), and tour length (TL). Through the memory function of the population, individual information is constantly updated and adjusted. Taking the connection graph, G, shown in Figure 5 as an example, if the algorithm starts when the ant is at the vertex, 1, the initialization information is:

NVV [1] = {1,2,3,4,5,6,7,8,9,10}.
VV [1] = {}.
NVE [1] = {(1,2),(3,4),(5,6),(7,8),(9,10)}
VE [1] = {}.
TL [1] = 0.0

After time, $\Delta t$, the pheromone on trajectory $(i, j)$ $i$ adjusted as follows:

$$\tau_{ij}(t + \Delta t) = \rho\tau_{ij}(t) + \Delta\tau_{ij} \tag{20}$$

where $\rho$ represents the volatilization rate of pheromone, $\tau_{ij}(t)$ represents the accumulation amount of pheromone on the track $(i, j)$ at time $t$, $\Delta\tau_{ij}$ represents the increment of the pheromone on the trajectory $(i, j)$ after the time, $\Delta t$, which can be calculated as follows:

$$\Delta\tau_{ij} = \sum_{k=1}^{m} \Delta\tau_{ij}^{k} \tag{21}$$

$\Delta\tau_{ij}^{k}$ denotes the pheromone on the trajectory $(i, j)$ during the searching process of the $k$-th ant, the expression of which is:

$$\Delta\tau_{ij}^{k} = \begin{cases} \frac{Q}{TL[k]}, & ant\ k\ pass\ path\ (i,\ j) \\ 0, & else \end{cases} \tag{22}$$

Among which, $Q$ is a constant. The pheromones on each trajectory during initialization are: $\Delta\tau_{ij} = 0$. At the time, $t$, the transition probability of an ant, $k$, from vertex $x$ to other feasible vertices is:

$$p_{ij}^{k} = \begin{cases} \frac{(\tau_{ij}(t))^{\alpha}(\eta_{ij}(t))^{\beta}}{\sum\limits_{s\in alowed_{k}} (\tau_{is}(t))^{\alpha}(\eta_{is}(t))^{\beta}}, & j \in alowed_{k} \\ 0, & otherwise \end{cases} \tag{23}$$

where $\eta_{ij}$ denotes the visibility on track $(i, j)$, which reflects the degree of heuristic from vertex $i$ to vertex $j$. Here, let $\eta_{ij} = 1/d_{ij}$, $d_{ij}$ is the distance from vertex $i$ to vertex $j$. The parameters, $\alpha$ and $\beta$, denote the influence weights of $\tau_{ij}(t)$ and $\eta_{ij}(t)$ on the whole transition probability. $alowed_{k}$ denotes the feasible neighborhood of the ant k at vertex i (end of the edge in the list NVE). Thus, the ant colony algorithm of the TTOI problem is shown in Figure 8.

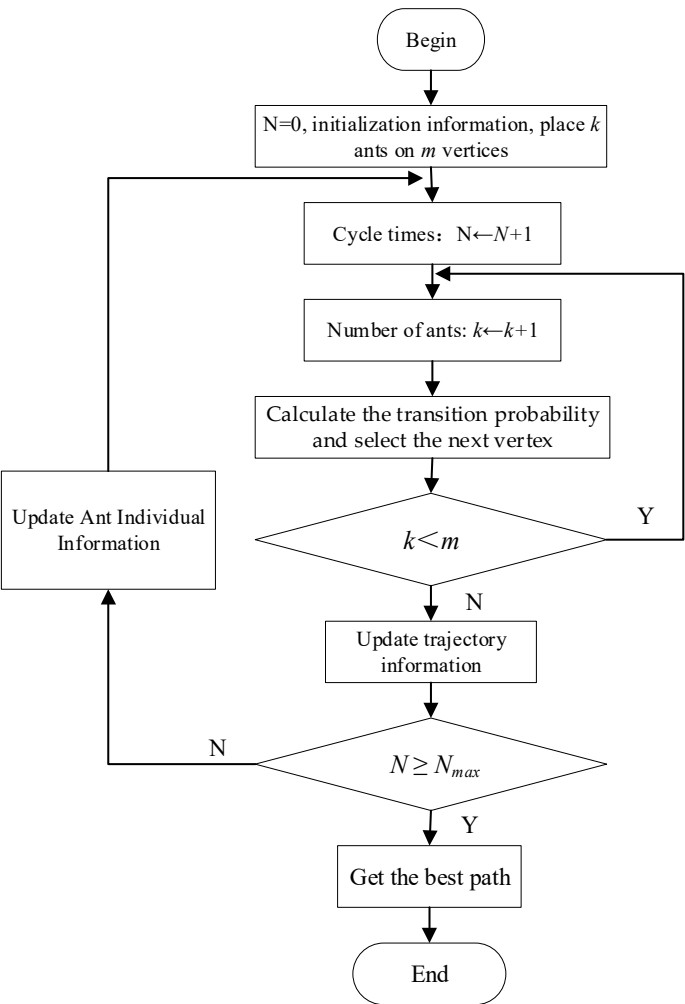

**Figure 8.** Flow chart of the ACO of the TTOI problem.

In the following, the validity of using ACO to solve the TTOI problem is verified by simulation experiments. Assuming that a 3D entity workpiece is divided into five patches, the number of edges in the connected graph, G, is five and the number of vertices is $m = 10$. The parameters of the algorithm are chosen as follows: $\alpha = 1$, $\beta = 5$, $\rho = 0.5$, $Q = 100$ and the maximum number of cycles is $N_{max} = 100$. Figure 9 shows the evolution of the optimal solution obtained from the algorithm. It can be seen from the figure that the length of the spray painting trajectory decreases monotonically with the evolutionary process. After about 70 generations of evolution, the length of the trajectory no longer changes and the algorithm converges.

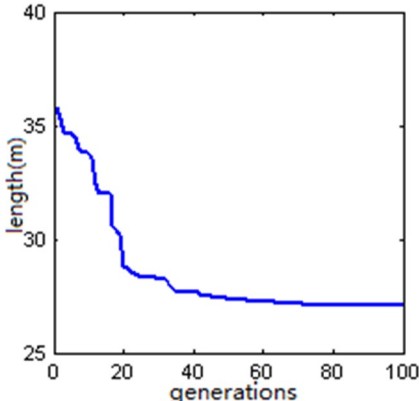

**Figure 9.** Simulation result of the ACO algorithm.

*5.4. Solving the TTOI Problem with Particle Swarm Optimization*

Particle swarm optimization (PSO) is easy to implement and there is no need to adjust a lot of parameters compared with other optimization algorithms. Also, it does not need gradient information, which is an effective tool to solve the optimal combination [17–19]. In the algorithm, each individual is a particle, and each particle represents a potential solution. Assuming that $z_i = (z_{i1}, z_{i2}, ..., z_{iD})$ is the D-dimensional position vector of the i-th particle, the current fitness value of $z_i$ can be calculated according to the fitness function so that the position of the particle can be measured. The process of calculating the minimum length of the spray path length can be selected as the fitness function in the TTOI problem. $v_i = (v_{i1}, v_{i2}, ..., v_{iD})$ is the flying speed of particle i; that is, the distance of particle movement. $p_i = (p_{i1}, p_{i2}, ..., p_{iD})$ is the optimal position of the particle to date. $p_g = (p_{g1}, p_{g2}, ..., p_{gD})$ is the optimal position of the particle swarm to date. In each iteration, the velocity and position of the particles can be updated according to:

$$v_{id}^{k+1} = v_{id}^k + c_1 r_1 (p_{id} - z_{id}^k) + c_2 r_2 (p_{gd} - z_{id}^k) \tag{24}$$

$$z_{id}^{k+1} = z_{id}^k + v_{id}^{k+1} \tag{25}$$

where $i = 1, 2, \ldots, m$, $d = 1, 2 \ldots D$, $r_1$ and $r_2$ are random numbers between [0, 1], and $c_1$ and $c_2$ are learning factors. Thus, the particle swarm algorithm of the TTOI problem is shown in the Figure 10.

Assuming that the 3D entity artifact is divided into five patches, the number of edges in the connection graph, G, shown in Figure 5 is five and the number of vertices is $m = 10$. To guarantee the precision of the algorithm, the maximum number of cycles is $N_{max} = 100$. To ensure that the particle does not skip the optimal solution and can search the search space sufficiently, let $\varepsilon = 1000$. To ensure accuracy and reduce the amount of calculation, take the number of particles as 20. The learning factors, $c_1$ and $c_2$, can make the particles have self-summary and the ability to learn from the outstanding individuals in the group, to be close to the best in their own history and the history within the group. These two parameters have little effect on the convergence of the algorithm, but if we adjust these two parameters properly, we can reach the convergence faster. After adjusting the values of $c_1$ and

$c_2$ several times and analyzing the effect of $c_1$ and $c_2$ on the optimal fitness, we can conclude that $c_1 = c_2 = 2$ is a better choice for the TTOI problem. Figure 11 shows the evolution of the optimal solution obtained from the algorithm. It can be seen from the figure that the length of the spray painting trajectory decreases monotonously with the evolutionary process, and finally tends to a definite value. As the spray painting robot does not consider the obstacle avoidance during the working process, the environment information is known and is relatively simple, so the particle swarm algorithm converges faster. It can be seen from Figure 11 that after about 80 generations of evolution, the length of the spray painting trajectory does not change and the algorithm converges.

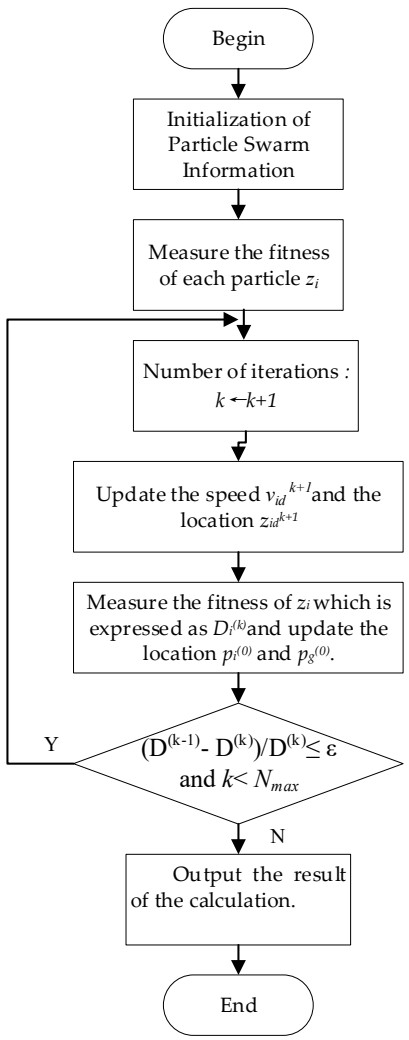

**Figure 10.** Flow chart of the PSO of the TTOI problem.

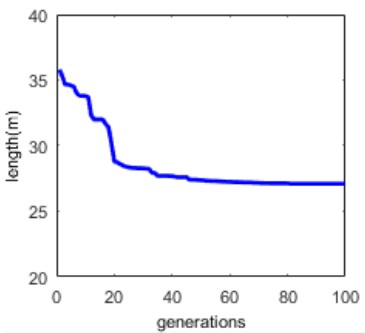

**Figure 11.** Simulation result of the PSO algorithm.

### 5.5. Comparison of Algorithms

In this paper, according to the built coating accumulation model: The finite range model, then FPAG (flat patch adjacency graph) is used to treat spraying workpiece, and the workpiece surface is divided into numerous patches. In each patch, the trajectory optimization algorithm in part 3 is used to obtain the trajectory. Finally, the genetic algorithm, ant colony algorithm, and particle swarm optimization algorithm are respectively used to obtain the optimal path to connect all patches to complete the spraying work. Assuming the ideal paint thickness of $q_d$ = 50 μm, the maximum allowable deviation of paint thickness of $q_w$ = 10 μm, the bottom radius of the cone paint sprayed by the gun is $R$ = 60 mm. The cumulative rate of the paint is obtained from the spray test data on the plate [12]:

$$f(r) = \frac{1}{15}(R^2 - r^2)\mathrm{\mu m/s} \tag{26}$$

After generating and optimizing the spray painting trajectories on the plate, the spray painting rate (at uniform speed) and the width of the overlapping area of the paint for each of the two spray strokes are obtained, which are $v$ = 256.3 mm/s, $d$ = 50.2 mm, respectively.

As shown in Figure 12, taking the experimental workpiece as an example, the workpiece can be regarded as a 3D entity. In the experiment, suppose the vertical distance from the gun to the workpiece surface is H, the bias algorithm can be used to obtain the spray path. Each spray parameter setting in the optimal algorithm of the spray painting trajectory are as follows: The ideal paint thickness is $q_d$ = 50 μm, the maximum allowable deviation of thickness is $q_w$ = ±10 μm, the spray radius is $R$ = 60 mm, the spray distance is $H$ = 100 mm, and the velocity of the spray is $v$ = 256.3 mm/s. The workpiece surface is divided into five patches after the modeling work and the path at the junction of the two patches is the PA-PA mode. GA, ACO, and PSO are used to carry out experiments when solving the TTOI. The parameters of the GA algorithm are set as: Population size is $P_{size}$ = 100, crossover probability is $x_{rate}$ = 0.20, mutation probability is $m_{rate}$ = 0.05, and the maximum number of evolutionary generation is $T$ = 100. The parameters of the ACO algorithm are set as: The number of ants is $m$ = 10, parameter $\alpha$ = 1, parameter $\beta$ = 5, the volatilization rate of the pheromone is $\rho$ = 0.5, constant $Q$ = 100, and the maximum number of iterations is $N_{max}$ = 100. The parameters of the PSO algorithm are set as: The maximum speed threshold is $\varepsilon$ = 1000, the number of particles is 20, learning factors are $c_1 = c_2 = 2$, and the maximum number of iterations is $N_{max}$ = 100.

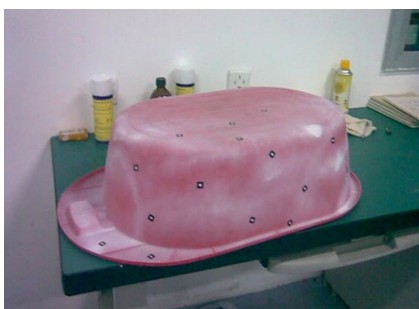

**Figure 12.** The experimental workpiece.

The self-developed offline programming system of the spray painting robot is used for the spray test [20,21]. The part of the optimized trajectories of different patches on the workpiece surface are shown in Figure 13. The paint thicknesses of 400 discrete points are measured by the paint thickness gauge after the spray painting operation. The paint thickness at the sample point after spray painting in the optimized trajectory is shown in Figure 14, among which the maximum paint thickness is $q_{max}$ = 55.6 μm, and the minimum paint thickness is $q_{min}$ = 45.5 μm. It can be seen that all the paint thicknesses at the sampling point are within the maximum allowable deviation of the paint thickness $q_w$, which is in line with the requirements of the spray quality.

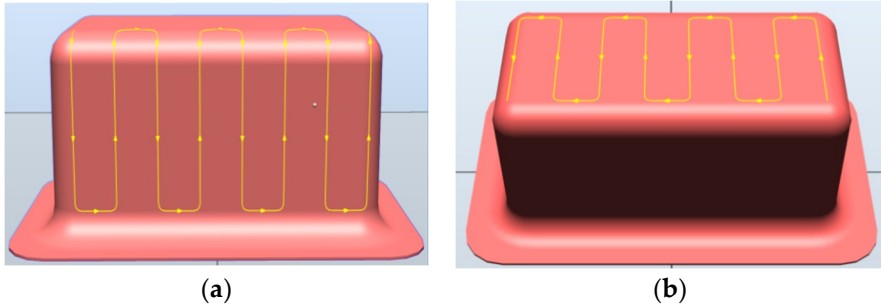

**Figure 13.** (**a**) Optimized trajectory at the side of the patch; (**b**) optimized trajectory at the top of the patch.

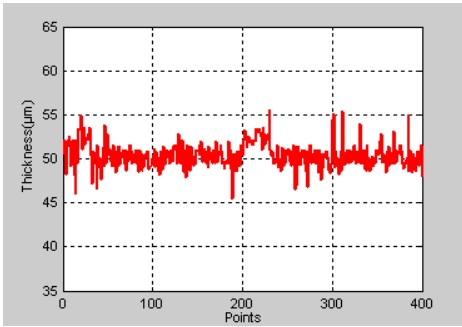

**Figure 14.** The paint thickness at the sampling points.

From the point of view of spray painting efficiency, the compared results of GA, ACO, PSO, and random combination are shown in Table 1. It can be seen from the results that the total length of the spray trajectory using the PSO algorithm is the shortest, the spray painting time is the least, and the execution time of the system operation is the longest, which is within the allowable range in the practical application. For the workpiece, the spray painting time was reduced by 23% compared to a random combination. The total trajectories of GA and ACO are shorter than those of the random combination, and the spray painting time are reduced by 16% and 20%, respectively. It should be noted that the workpiece used here is only divided into five patches. For the workpiece, which is more complex and has more patches, the advantages of using the PSO algorithm in saving painting time will be more obvious, but the execution time of the offline programming system will be longer. Therefore, if the real-time performance of the system operations can meet the practical application requirements, the PSO algorithm will be the best choice. Otherwise, the GA algorithm or ACO algorithm can be taken into consideration.

**Table 1.** The compared results of different algorithms.

|  | GA | ACO | PSO | Random Combination |
|---|---|---|---|---|
| Total length of Spray Path (*m*) | 28.4 | 27.6 | 26.8 | 29.8 |
| Spray painting Time of Robot (*s*) | 94 | 89 | 86 | 112 |
| Execution Time of Operation (*s*) | 0.23 | 0.35 | 0.52 | 0.10 |

*5.6. Spray Painting Experiment*

As shown in Figures 15–18, taking the automotive body of a brand is as the paint objective. The thickness of the coating on each sampling point is shown in Figure 19. Each spray parameter settings in the optimal algorithm of the spray painting trajectory are as follows: The ideal paint thickness is $q_d = 50$ μm, the maximum allowable deviation of the thickness is $q_w = \pm 10$ μm, the spray radius is $R = 60$ mm, the spray distance is $H = 80$ mm, and the velocity of the spray is $v = 389$ mm/s. The left side of the car surface is divided into 15 patches and the trajectory of each patch is in a different

color. GA, ACO, and PSO are used to carry out experiments when solving the TTOI. The parameters of the GA algorithm are set as: Population size is $P_{size} = 100$, crossover probability is $x_{rate} = 0.20$, mutation probability is $m_{rate} = 0.05$, and the maximum number of evolutionary generation is $T = 200$. The parameters of the ACO algorithm are set as: The number of ants is $m = 10$, parameter $\alpha = 1$, parameter $\beta = 5$, the volatilization rate of pheromone is $\rho = 0.5$, constant $Q = 100$, and the maximum number of iterations is $N_{max} = 200$. The parameters of the PSO algorithm are set as: The maximum speed threshold is $\varepsilon = 1000$, the number of particles is 20, learning factors $c_1 = c_2 = 2$, and the maximum number of iterations is $N_{max} = 200$.

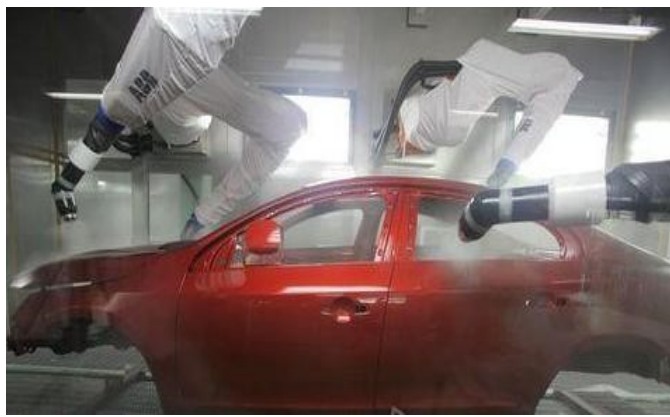

**Figure 15.** An automobile paint line composed by several spray-painting robots.

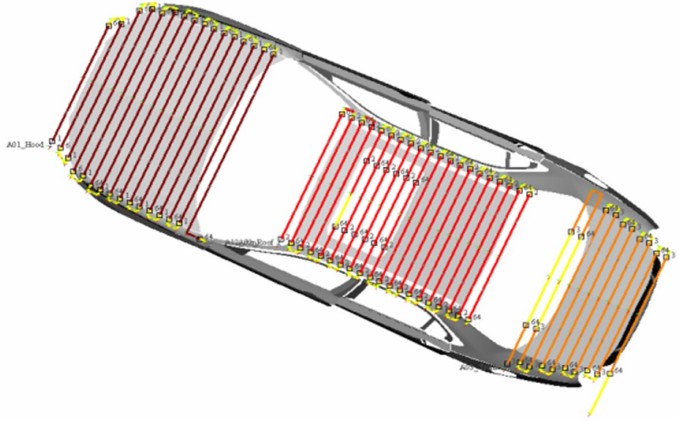

**Figure 16.** Spray painting trajectory on the roof part of the automobile.

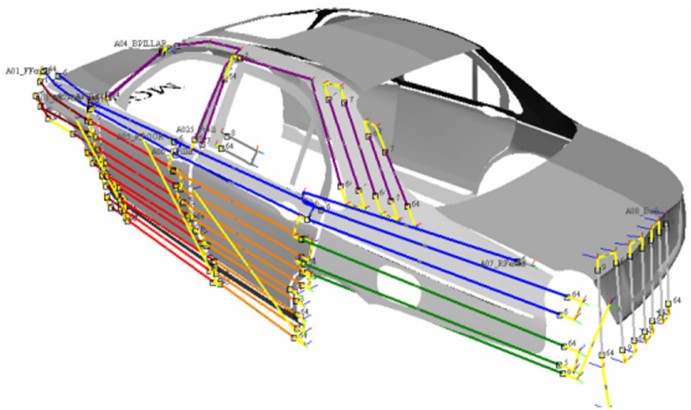

**Figure 17.** Spray painting trajectory on the left part and the left rear part of the automobile.

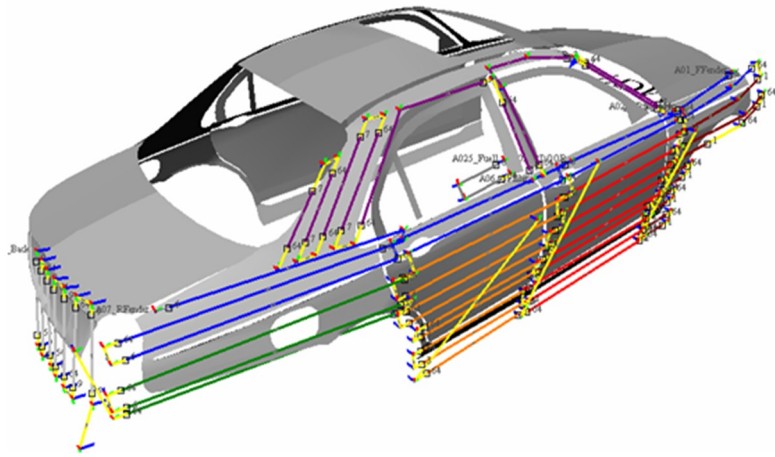

**Figure 18.** Spray painting trajectory on the right part and the right rear part of the automobile.

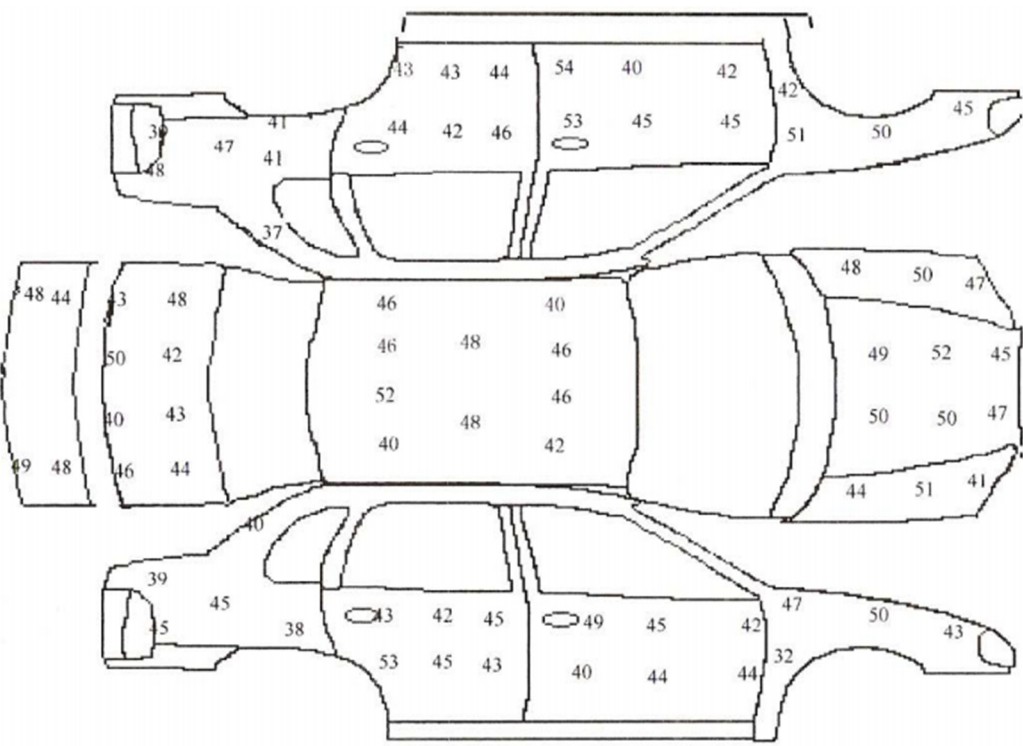

**Figure 19.** The paint thickness at the sampling points on the automobile body.

From the point of view of spray painting efficiency, the compared results of the GA, ACO, PSO, and random combination are shown in Table 2. It can be seen from the results that the total length of the spray trajectory using the PSO algorithm is the shortest, the spray painting time is the least, and the execution time of the system operation is the longest, which is within the allowable range in the practical application.

**Table 2.** The compared results of different algorithms.

|  | GA | ACO | PSO |
|---|---|---|---|
| Total length of Spray Path (*m*) | 124.2 | 116.9 | 103.6 |
| Spray painting Time of Robot (*s*) | 282 | 270 | 253 |
| Execution Time of Operation (*s*) | 0.9 | 1.1 | 1.5 |

## 6. Conclusions

In this paper, the spatial trajectory optimization method of a spray painting robot for 3D entity objects was proposed. Firstly, the finite range model of the paint deposition rate was established, and the 3D entity was sliced by the surface modeling method according to FPAG. Then, after planning the spray path on each patch, the variance of the paint thickness of the discrete point and the ideal paint thickness was taken as the objective function and the trajectory on each patch was optimized. The path at the junction of two patches was the PA-PA (parallel-parallel) mode. The improved GA algorithm, ACO algorithm, and PSO algorithm were used to solve the TTOI problem. The practicability of the algorithms was verified by simulation experiments. Finally, spraying experiments were conducted on the off-line programming experimental platform of the spraying robot, and the results of the three algorithms were studied. The results of the automobile body spraying experiments showed that the proposed trajectory optimization of the 3D entity spraying robot can completely satisfy the requirements of uniformity of the spraying thickness. More experimental data please see the Supplementary Materials.

**Supplementary Materials:** The following are available online at http://www.mdpi.com/2079-9292/8/1/74/s1.

**Author Contributions:** W.C. and Y.T. conceived and designed the experiments; W.C. performed the experiments; H.L. and J.L. analyzed the data; X.W. contributed reagents/materials/ analyzed the data; X.W. contributed reagents/materials/analysis tools; X.W. wrote the paper.

**Funding:** This research was supported by the National Natural Science Foundation of China under grant 61503162, "Six talent peaks" high-level talents projection Jiangsu Province under grant 2016GDZB021, Project funded by China Postdoctoral Science Foundation under grant 2016M601691, Industrial Foresight Project of Zhenjiang City under grant GY2018018.

**Conflicts of Interest:** The authors declare no conflict of interest.

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
