# Peer review of "Optimized Combination of Spray Painting Trajectory on 3D Entities"

_electronics, doi:10.3390/electronics8010074_

Round 1
Reviewer 1 Report
Well written paper. Some experimental results to validate the film build consistency over the surface of different body styles would have been extremely helpful.
Author Response
Dear Editors and Reviewers:
Thank you for your letter concerning our manuscript entitled “Optimized Combination of Spray Painting Trajectory on 3D Entities” (ID: electronics-405372). Thank you very much for your affirmation of my research. I will continue to work in this direction.
Special thanks to you for your good comments.

Reviewer 2 Report
In this paper, authors presented the optimization of spatial trajectory for spray painting robot for 3D objects. Authors used tools like Genetic algorithm (GA), Ant colony optimization (ACO) and Particle swarm optimization (PSO) algorithms to solve the tool trajectory optimal integration problem. Authors have used several steps to optimize the robotic paint spraying process for 3D entities. For example, they used segmentation methods using FPAG of a 3D object by dividing them into small patches, then optimized the trajectory for each patch and used different optimization algorithms to find the optimal combination of trajectories for TTOI problem. Experimental results suggest that PSO gives best optimal trajectory in terms of spraying time and the total length of the spray trajectory. The novelty of the work mainly lies in the comparison between the algorithms. However, individual algorithms have already been applied in the different robotic trajectory optimization processes. In addition, the reviewer found several issues in the manuscript. Please find it as following
1. Paper is written very badly, there are a lot of grammatical, comma and space issues.
• For example, line 42 Figure should be written in the same manner and should be followed the journal’s format.
• Line 52, it should be mathematical, line 71, 79, 96, 101, 102, 132, 140, 195 etc. There are a lot of spacing issues, please revise the whole manuscript.
2. Reviewer still does not convince the novelty of the work, since the individual algorithms are already been applied for different robotic trajectory optimization. It would be great if authors can highlight their novel contribution with more details.
3. The introduction seems very short, several related works that needs to be added.
4. In section 2, what is c(θ,φ), Px etc. is not defined in the text.
5. Reviewer wonders, why the amount of paint equation does not include any time of spraying and rate of spraying, reviewer believes that spraying longer time and the higher rate will increase the paint amount rather than just angle of spray and distance.
6. Figure 2 should be redrawn to avoid the text and line interference.
7. Line 94 equation 4.4 is not mentioned in the text.
8. Reviewer also suggests the formatting of equation numbers are very irregular throughout the manuscript.
9. There several symbols used in the equations but not mentioned in the text or caption, please revise it throughout the manuscript. Such as R and d is not mentioned, and it is difficult to understand.
10. Reviewer suggests that follow the journal formatting for figure captioning and it should not necessary if all the words should start from the capital.
11. Acronyms are not the explained like TCP, please check throughout the manuscript, add in abstract too.
12. In the line 159 used type of comma is wrong, there are many these kinds of error throughout the manuscript. Please revise it.
13. It seems Figure 4 and Figure 5 are taken from different paper and not cited in the text and caption, please modify Figure 5 to avoid line and symbol intersection.
14. In line 194 citation is missing
15. Reviewer suggests that from line 310 to 321 should be a flowchart and should be explained better, similarly lines from 347 to 355 should be a flowchart
16. Reviewer doubts on Figure 8, It seems not conversing, it would be interesting to check it for more than 100 generations
17. It seems equation 26 has been taken from authors previous work without any mention and citation in the text.
18. It seems all values of v, d, qd, qm are already mentioned in the authors' previous articles, reviewer wonders, if it is new experiments or taken from the previous experiments, if yes please add the citation. Reviewer suggest lines from 390 to 396 should remove, because they already mentioned in the previous sections.
19. It is the same doubt if the Figure 9 and Figure 11 are new or previous work, please check and add the citations.
20. Unit of the variables should be uniform throughout the text, it is like authors used some places µm and some places 10-6 m. Please revise it throughout the manuscript.
21. What is statistical significance of the data from the table 1 and table 2
22. It is again putting reviewer on doubt, how the Figure 13, 14, 15 and 16 are related to this work. These images are also mentioned in the previous article of the authors. Please explain.
23. Reviewer is also interested in how authors are relating the experimental work and simulation work. It is recommended to add more details in script.
24. It is desired to add the experimental method in detail and relate the simulation work.
25. Authors are highly recommended to revise whole manuscript with formatting of space, comma, figure caption, equation caption, proper citations, equations numbers in, subscript and superscript etc.
26. Reviewer is recommending to a video of their experimental work.
Author Response
Dear Editors and Reviewers:
Thank you for your letter and for the reviewers’ comments concerning our manuscript entitled “Optimized Combination of Spray Painting Trajectory on 3D Entities” (ID: electronics-405372). Those comments are all valuable and very helpful for revising and improving our paper, as well as the important guiding significance to our researches. We have studied these comments carefully and have made corrections which we hope meet with approval. The main corrections in the paper and the responses to the reviewer’s comments are in the attachment. Thank you for your careful reading of the responses.
Special thanks to you for your good comments.

Round 2
Reviewer 2 Report
Reviewer is convinced with authors rebuttal, it seems manuscript can go for publication after these minor changes.
1) Please add the point 23 in the text.
2) It seems Figure 15 and Figure 16 captions are flipped
3) Please correct spelling mistakes in the caption of Figure 18
4) Add the attachment part of rebuttal as supplementary information, reviewer believes, it would be useful information for readers.
Author Response
Dear Editors and Reviewers:
Thank you for your letter and for the reviewers’ comments concerning our manuscript entitled “Optimized Combination of Spray Painting Trajectory on 3D Entities” (ID: electronics-405372). Those comments are all valuable and very helpful for revising and improving our paper, as well as the important guiding significance to our researches. We have studied these comments carefully and have made corrections which we hope meet with approval. The main corrections in the paper and the responses to the reviewer’s comments are in the attachment.
